# LEARNING KOLMOGOROV MODELS FOR BINARY RANDOM VARIABLES

## ABSTRACT

We propose a framework for learning a *Kolmogorov model*, for a collection of binary random variables. More specifically, we derive conditions that *link* (in the sense of implications in mathematical logic) outcomes of specific random variables and extract valuable relations from the data. We also propose an efficient algorithm for computing the model and show its *first-order optimality*, despite the combinatorial nature of the learning problem. We exemplify our general framework to recommendation systems and gene expression data. We believe that the work is a significant step toward interpretable machine learning.

## 1 INTRODUCTION

Machine learning and artificial intelligence method have permeated a large number of areas (Marr, Sept 2016). These methods are based on *machine learning models*, which consist of learning an input-output mapping for a given dataset. Despite the plethora of such models (e.g., matrix factorization (Koren et al., 2009), SVD-based models (Koren, 2008), deep neural networks (LeCun et al., 2015), and models inspired from physics (Stark, 2016b)), they lack *interpretability*: they are not capable of offering insight about the data, nor the underlying process. The lack of interpretablity may have serious consequences in mission-critical systems, ethics, and validation of computer-aided diagnosis (Doshi-Velez & Kim, 2017). While there is no consensus around the definition of interpretability, *causality* (Lipton, 2016) is a vital component: it refers to *causal relations* within the data and insight about the underlying data-generating process. We thus adopt a generalized version of causality using *implication* in mathematical logic as our 'definition' of *interpretable models*.

To this end, we propose learning a so-called *Kolmogorov Model (KM)* associated with a set of binary Random Variables (RVs). In addition to prediction, the interpretability of the model (as defined above) enables learning *causal relations*.[1] We derive a sufficient condition under which the realization of one RV's outcome (causally) *implies* the outcome of the other. In the context of recommendation systems, causal relations identify groups of items, for which a user liking one item *implies* that he/she likes all other items in the group. In gene expression analysis, the same relations identify groups of DNA locations for which the expression of a gene in one of them implies its expression in all other locations. The foundation of our approach is to model binary RVs as elementary events on a Kolmogorov space, by an inner product of two vectors (formally stated in Section 2.1), which is based on established results from classical probability theory. To our best knowledge, this specific formulation is novel in the context of learning representation.

The inner product of our formulation is reminiscent of factorization methods such as, matrix factorization (Koren, 2008), non-negative matrix factorization (Lee & Seung, 2001), SVD (Cai et al., 2010), and physics-inspired techniques, non-negative models (Stark, 2016a). It should be noted that the inner product components of these methods is usually based on strong intuition about the data, provided and validated by a human expert. In contrast, the proposed KM is a fully automated approach, deeply rooted in probability theory, which finds an interpretable model (as defined above) of the data without any human intervention. Thus, both the model and the causal relations within the data have a strong mathematical basis. Moreover, in most of the existing approaches, we may need to have one pipeline for learning the representation and another one for mining these relations. However, our approach requires only a single pipeline for both tasks. This is very appealing on a

---

[1]Hereafter, for the sake of simplicity, causal relations mean logical relations within the data.

conceptual level, and it can drastically simplify coding and validation. Our method also generalizes K-means and some of its variants. Detailed discussions of the relation between our proposed method and these prior works is in Appendix A.3.

In an abstract sense, we formulate a KM learning problem as a *coupled combinatorial program*, decompose it into two subproblems using the *Block-Coordinate Descent (BCD)* method, and obtain provably optimal solutions for both. For the first one, we exploit the structure of linear programs on the unit simplex and use low-complexity (yet optimal) *Frank-Wolfe* algorithm (Frank & Wolfe, 1956). To bypass the inherent complexity of the second subproblem (combinatorial and NP-hard), we propose a semidefinite relaxation, and show its *quasi-optimality* in recovering the optimal solution of the combinatorial subproblem. Finally, we show the convergence of our algorithm to a stationary point of the original problem. We propose a simple algorithm for mining the causal relations. All the proofs and additional discussions are available in Authors (Oct 2017).

## 2 SYSTEM MODEL

Notation: Lowercase letter $a$, uppercase bold letter $\boldsymbol{A}$, and calligraphic letter $\mathcal{A}$ denote vectors, matrices, and sets, respectively. $[\boldsymbol{A}]_{i,j}$ and $\boldsymbol{A}^T$ denote element $(i, j)$ and the transpose of $\boldsymbol{A}$. $\mathrm{supp}(\boldsymbol{a})$ denotes the support set of $\boldsymbol{a}$. The inequality $\boldsymbol{x} \leq \boldsymbol{y}$ holds element-wise. $\boldsymbol{I}$ denotes the identity matrix, $\boldsymbol{1}$ and $\boldsymbol{0}$ the all-one and all-zero vectors (of appropriate dimension). $\boldsymbol{e}_n$ is the $n$th elementary basis, $\mathcal{P} = \{\boldsymbol{p} \in \mathbb{R}^D_+ \mid \boldsymbol{1}^T \boldsymbol{p} = 1\}$ the unit simplex, and $\{n\} := \{1, \cdots, n\}$.

### 2.1 PROBLEM FORMULATION

Consider a double-indexed set of binary *Random Variables (RVs)*, $X_{u,i} \in \mathcal{A} = \{1, 2\}$, taken from a dataset $\mathcal{D} = \{(u, i) \mid (u, i) \in \mathcal{U} \times \mathcal{I}\}$. Each RV is defined on a sample space $\Omega$, consisting of *elementary events* $\Omega = \{\omega_d \mid 1 \leq d \leq D\}$. We denote by $\mathbb{P}[X_{u,i} = z]$, $z \in \mathcal{A}$, the probability that RV $X_{u,i}$ takes the value $z \in \mathcal{A}$; see example in Section 2.2. Since $X_{u,i}$ is binary, it is fully characterized by considering one outcome. Thus, we write the *Kolmogorov Model (KM)* for $X_{u,i}$ as

$$\text{KM:} \qquad \mathbb{P}[X_{u,i} = 1] = \boldsymbol{\theta}_u^T \boldsymbol{\psi}_i \ . \tag{1}$$

Thus, each RV $X_{u,i}$ is associated with (characterized by) a *Probability Mass Function (PMF)* $\boldsymbol{\theta}_u, u \in \mathcal{U}$, and an *indicator vector* $\boldsymbol{\psi}_i, i \in \mathcal{I}$. The model follows from established results in classical probability theory (Kolmogorov, 1957),(Gray, 2009) (formalized in Appendix A.2). Notice that the model in (1) can approximate with arbitrarily small accuracy the measure corresponding to $\mathbb{P}[\cdot]$ for a large enough $D$.

**Problem 1 (Problem Statement)** Let $p_{u,i}$ denote the *empirical value* of $\mathbb{P}[X_{u,i} = 1]$. We assume that $\{p_{u,i}\}$ are known for elements of a *training set* $\mathcal{K} \subseteq \mathcal{D}$, where $\mathcal{K} = \{(u, i) \mid (u, i) \in \mathcal{U} \times \mathcal{I}\}$.[2] Given samples coming from the model in (1), we wish to infer the parameters of underlying probability distribution: find parameters of the KM, i.e., $\{\boldsymbol{\psi}_i, \boldsymbol{\theta}_u\}$ that best describe $\{p_{u,i} \mid (u, i) \in \mathcal{K}\}$. The resulting problem is a *fully parametric statistical inference* task. For tractability, we address it using the minimum mean-squared error as a point estimator. The corresponding optimization problem is

$$(Q): \quad \{\boldsymbol{\psi}_i^\star, \boldsymbol{\theta}_u^\star\} \in \begin{cases} \underset{\{\boldsymbol{\psi}_i\}, \{\boldsymbol{\theta}_u\}}{\mathrm{argmin}} \sum_{(u,i) \in \mathcal{K}} \left(\boldsymbol{\theta}_u^T \boldsymbol{\psi}_i - p_{u,i}\right)^2 \triangleq \mathcal{E}(\{\boldsymbol{\psi}_i\}, \{\boldsymbol{\theta}_u\}) \\ \text{s. t. } \boldsymbol{\theta}_u \in \mathcal{P} \ , \ \boldsymbol{\psi}_i \in \mathbb{B}^D, \ \forall (u, i) \in \mathcal{K} \end{cases} . \tag{2}$$

We present our solution to this *coupled non-convex conspiratorial* optimization in the next section. The obtained solution to $(Q)$ can be used for prediction on a test set, as well as extracting causality structures within training set (see Section 4).

**Proposed Approach:** While the proposal to model binary RVs as elementary events on a Kolmogorov space is based on established results, the specific learning formulation (Problem 1) is novel. Because this model is rooted in probability theory, (1) defines the outcome of a RV in the *strict* Kolmogorov sense, and the resulting causal relations (Section 4) also hold from a strict analytical perspective. Note that causal relations (a.k.a. *association rules*) may still be extracted using existing methods,

---

[2]The method for acquiring (estimates of) the empirical probabilities is application-dependent (see Section 5).

e.g., (non-negative) Matrix Factorization (MF) and their variants, SVD, binary MF, and K-means. However, these relations are not based on causality and the formal relations that (mathematically) follow from the KM in (1), but rather on intuitions/heuristics, which may yield different relations. Naturally, we wish the explore further in this work the causal relations that arise from the proposed model. Additionally, unlike existing methods, the prediction and causal relations mining are done in 'one-shot', thereby simplifying the implementation/validation; see Appendix A.3.

## 2.2 Illustrative Example: Recommendation Systems

In this context, $\mathcal{U}$ and $\mathcal{I}$ denote the set of users and items respectively, and $X_{u,i}$ models the preference of user $u$ for item $i$, $(u,i) \in \mathcal{K}$. Thus, $\mathbb{P}[X_{u,i} = 1]$ (or $\mathbb{P}[X_{u,i} = 2]$) is the probability that user $u$ likes (or dislikes) item $i$. Moreover, $\boldsymbol{\theta}_u$ determines the profile/taste of user $u$, $\boldsymbol{\psi}_i$ is related to item $i$ (depending on genre, price, etc.), and the elementary events denote movie genres (e.g., $\omega_1 =$ "Action", $\omega_2 =$ "SciFi", etc.). The *training set*, consisting of an *empirical probability* that user $u$ likes item $i$, $p_{u,i}$. We obtain this probability using $p_{u,i} \triangleq [\boldsymbol{R}]_{u,i}/R_{\max}$, where $[\boldsymbol{R}]_{u,i} \in \mathbb{N}$ denotes the rating that user $u$ has provided for item $i$, and $R_{\max}$ the maximum rating (Stark, 2015). For instance, if user $u$ rates item $i$ with a score of $[\boldsymbol{R}]_{u,i} = 7$ (where the maximum rating is 10), then $p_{u,i} = 0.7$; Other approaches may be used to obtain $p_{u,i}$ depending on the specific application.

As a concrete illustrative example, consider a 10-star "recommendation system", having 2 users and 2 items. We then find the $D$-dimensional ($D = 3$) KM factorization to obtain, $\{\boldsymbol{\psi}_i^\star\}_{i=1}^2$ and $\{\boldsymbol{\theta}_u^\star\}_{u=1}^2$. $D$ is the size of the Kolmogorov space $\Omega$, the number of elementary events, and the dimension of the factorization (selected via cross-validation to minimize the test error). Solving $(Q)$ results in finding $\{\boldsymbol{\psi}_i^\star\}_{i=1}^2$ and $\{\boldsymbol{\theta}_u^\star\}_{u=1}^2$. An example result is given below:

$$\underbrace{\begin{bmatrix} 0.3 & 1 \\ 0.1 & 1 \end{bmatrix}}_{\{p_{u,i}\}} = \begin{bmatrix} \boldsymbol{\theta}_1^{\star T} \begin{Bmatrix} 0.2 & 0.3 & 0.5 \end{Bmatrix} \\ \boldsymbol{\theta}_2^{\star T} \begin{Bmatrix} 0.1 & 0.1 & 0.8 \end{Bmatrix} \end{bmatrix} \begin{bmatrix} 0 & 1 & \text{\}Action} \\ 1 & 1 & \text{\}SciFi} \\ \underbrace{0}_{\boldsymbol{\psi}_1^\star} & \underbrace{1}_{\boldsymbol{\psi}_2^\star} & \text{\}Drama} \end{bmatrix}$$

To showcase the model's intuition, note that $p_{1,1}$, the probability that user 1 likes movie 1, is represented as $\boldsymbol{\psi}_1^T \boldsymbol{\theta}_1$. It is thus expressed as *convex/stochastic* mixture of *movie genres*, since elementary events are movie genres in this scenario. We underline that this high degree of interpretability is *not artificially* enforced but rather context-dependent. More generally, a KM represents a set of observed outcomes for RVs as (context-dependent) mixtures of elementary events. The approach consists of learning a (hidden) latent model by jointly optimizing the PMF and binary indicator vectors. Thus, interpreting elements of the indicator vectors as movie genres requires a *context-dependent map*, from the elements of $\boldsymbol{\psi}_i^\star$ to movie genres. Another way to find such an interpretation is when this context-dependent map is known apriori. In this setting, each item is already tagged with its movie genres, and $\boldsymbol{\psi}_i$ need not be optimized; Alas, having this context-dependent map comes at the expense of a loss in training/test (see Appendix A.4). However, recall that the primary interest of this work is not this facet of interpretability, but rather by that of the causal relations.

## 3 Proposed Algorithm

To approach a solution for problem (2), we use the *block-coordinate descent (BCD)* method to split $(Q)$ into two sub-problems. Here, we derive our solution approach to each. We first refine the current PMF estimation $\boldsymbol{\theta}_u$, and then that of the indicator $\boldsymbol{\psi}_i$. Given $\{\boldsymbol{\psi}_i^{(n)}\}$ at iteration $n$, we pose the PMF refinement, $\theta$-step, as

$$(Q_1) : \boldsymbol{\theta}_u^{(n+1)} \in \underset{\boldsymbol{\theta}_u \in \mathcal{P}}{\operatorname{argmin}} \, f(\boldsymbol{\theta}_u) \triangleq \boldsymbol{\theta}_u^T \underbrace{\boldsymbol{Q}_u^{(n)}}_{:=\sum_{i \in \mathcal{I}_K} \boldsymbol{\psi}_i^{(n)} \boldsymbol{\psi}_i^{(n)T}} \boldsymbol{\theta}_u - 2\boldsymbol{\theta}_u^T \underbrace{\boldsymbol{r}_u^{(n)}}_{:=\sum_{i \in \mathcal{I}_K} \boldsymbol{\psi}_i^{(n)} p_{u,i}} . \quad (3)$$

We then pose the indicator vector refinement, $\psi$-step, as

$$(Q_2) : \boldsymbol{\psi}_i^{(n+1)} \in \underset{\boldsymbol{\psi}_i \in \mathbb{B}^D}{\operatorname{argmin}} \, g(\boldsymbol{\psi}_i) \triangleq \boldsymbol{\psi}_i^T \underbrace{\boldsymbol{S}_i^{(n+1)}}_{:=\sum_{u \in \mathcal{U}_K} \boldsymbol{\theta}_u^{(n+1)} \boldsymbol{\theta}_u^{(n+1)T}} \boldsymbol{\psi}_i - 2\boldsymbol{\psi}_i^T \underbrace{\boldsymbol{v}_i^{(n+1)}}_{:=\sum_{u \in \mathcal{U}_K} \boldsymbol{\theta}_u^{(n+1)} p_{u,i}} . \quad (4)$$

**function** $[\boldsymbol{\theta}_u^\star]$ = FW ( $\boldsymbol{Q}_u, \boldsymbol{r}_u, \epsilon$ )
  **for** $k = 1, 2, ..., I_{FW}$ **do**
    $\boldsymbol{d}_u^{(k)} = \boldsymbol{e}_{j^\star}$, $j^\star = \underset{1 \le j \le D}{\operatorname{argmin}} \, [\nabla f(\boldsymbol{\theta}_u^{(k)})]_j$
    $\boldsymbol{\theta}_u^{(k+1)} = (1 - \alpha_u^{(k)})\boldsymbol{\theta}_u^{(k)} + \alpha_u^{(k)}\boldsymbol{d}_u^{(k)}$
    Stop if $\|\boldsymbol{\theta}_u^{(k+1)} - \boldsymbol{\theta}_u^{(k)}\| \le \epsilon$
  **end for**
**end function**

Table 1: $\theta$-step solution using FW

**function** $[\hat{\boldsymbol{\psi}}_i]$ = SDR ( $\boldsymbol{S}_i, \boldsymbol{t}_i, M_{rnd}$ )
  *// Repeat to approximate each $\boldsymbol{\psi}_i^\star$, $\forall i \in \mathcal{I}_K$*
  Solve (5) to find $\boldsymbol{X}_i^{(\text{SDR})}$
  Factorize as $\boldsymbol{X}_i^{(\text{SDR})} = \boldsymbol{L}_i^T \boldsymbol{L}_i$
  **for** $m = 1, 2, ..., M_{rnd}$ **do**
    Gen. zero-mean i.i.d Gaussian vector $\boldsymbol{u}_i^{(m)}$
    Compute $\hat{\boldsymbol{u}}_i^{(m)} = \text{sign}[\, \boldsymbol{L}_i^T \boldsymbol{u}_i^{(m)} \,]$
  **end for**
  Find $m^\star = \operatorname{argmin}_{1 \le m \le D+1} \, \hat{\boldsymbol{u}}_i^{(m)^T} \tilde{\boldsymbol{S}}_i \hat{\boldsymbol{u}}_i^{(m)}$
  Compute $\hat{\boldsymbol{z}}_i = [\boldsymbol{u}_i^{(m^\star)}]_{1:D} \, [\boldsymbol{u}_i^{(m^\star)}]_{D+1}$
  Approximate $\boldsymbol{\psi}_i^\star$, as $\hat{\boldsymbol{\psi}}_i = (\hat{\boldsymbol{z}}_i + \mathbf{1})/2$
**end function**

Table 2: $\psi$-step solution using SDR

Moreover, $\mathcal{U}_K$ and $\mathcal{I}_K$ are defined as $\mathcal{K} = \{(u, i) \mid u \in \mathcal{U}_K \subseteq \mathcal{U} \,, \, i \in \mathcal{I}_K \subseteq \mathcal{I}\}$. Recall that having *globally optimal* solutions for both $(Q_1)$ and $(Q_2)$ is necessary to show the convergence of BCD (Tseng, 2001) - a challenging task due to the NP-hardness of $(Q_2)$.

### 3.1 $\theta$-STEP: REFINE PMF ESTIMATE

We use the *Frank-Wolfe (FW)* algorithm (Frank & Wolfe, 1956) to solve $(Q_1)$ as a succession of Linear Programs (LPs) over the unit simplex. While LP solvers generally have similar complexity as quadratic program solvers, solving an LP reduces to searching for the minimum index - which is computationally efficient, when the LP is over the unit simplex. We summarized this FW variant for solving $(Q_1)$; see also Jaggi (2013)[Algorithm 1]. Here, we drop the BCD iteration index, $n$, and just keep the FW iteration number, $k$, for notation simplicity. We first determine the *descent direction*: $\boldsymbol{d}_u^{(k)} \in \operatorname{argmin}_{\boldsymbol{s} \in \mathcal{P}} \left( \nabla f(\boldsymbol{\theta}_u^{(k)}) \right)^T \boldsymbol{s}$. The constraint $\boldsymbol{s} \in \mathcal{P}$ greatly simplifies the above LP and yields: $\boldsymbol{d}_u^{(k)} = \boldsymbol{e}_{j^\star}$, $j^\star \in \operatorname{argmin}_{1 \le j \le D} \, [\nabla f(\boldsymbol{\theta}_u^{(k)})]_j$ ; see Proposition 2 (Appendix A.6). The solution follows from LPs over the unit probability simplex. Thus, finding the descent direction reduces to searching over the $D$-dimensional gradient vector (done in $\mathcal{O}(D)$). Then, the current value is updated using a simple step size rule, $\alpha_u^{(k)} = k/(k+1)$. Table 1 shows the $\theta$-step solution, and Proposition 3 in Appendix A.6 characterizes its convergence.

### 3.2 $\psi$-STEP: REFINE INDICATOR ESTIMATE

To address the NP-hard nature of $(Q_2)$, we propose a solution based on *Semi-Definite Relaxation and Randomization (SDR)*, and establish its *quasi-optimality*. We use the results of Ma et al. (2002)[Sec IV-C]) and a series of reformulations to rewrite $(Q_2)$ in the following equivalent form (see Authors (Oct 2017) for all the derivations):

$$\boldsymbol{X}_i^\star \in \operatorname{argmin}_{\boldsymbol{X}_i} \text{tr}(\tilde{\boldsymbol{S}}_i \boldsymbol{X}_i) \,, \quad \text{s. t. } \boldsymbol{X}_i \succeq 0, \, [\boldsymbol{X}_i]_{k,k} = 1, \forall k \,, \, \text{rank}(\boldsymbol{X}_i) = 1$$

where $\boldsymbol{X}_i = \boldsymbol{x}_i \boldsymbol{x}_i^T$, $\tilde{\boldsymbol{S}}_i = \begin{bmatrix} (1/4)\boldsymbol{S}_i & -\tilde{\boldsymbol{t}}_i/2 \\ -\tilde{\boldsymbol{t}}_i^T/2 & 0 \end{bmatrix}$, $\boldsymbol{x}_i = \begin{bmatrix} \boldsymbol{z}_i \\ w_i \end{bmatrix}$, $\boldsymbol{z}_i = 2\boldsymbol{\psi}_i - \mathbf{1}$, $w_i \in \{-1, +1\}$ is an auxiliary variable, and $\tilde{\boldsymbol{t}}_i \triangleq (\boldsymbol{v}_i - (1/2)\boldsymbol{S}_i \mathbf{1})$. The problem is then relaxed into a convex program,

$$\boldsymbol{X}_i^{(\text{SDR})} \in \operatorname{argmin}_{\boldsymbol{X}_i} \text{tr}(\tilde{\boldsymbol{S}}_i \boldsymbol{X}_i) \,, \quad \text{s. t. } \boldsymbol{X}_i \succeq 0, \, [\boldsymbol{X}_i]_{k,k} = 1, \forall k \tag{5}$$

$\boldsymbol{X}_i^{(\text{SDR})}$ may be solved using generic solvers. Then, a randomization procedure (Ma et al., 2002) extracts an approximate (binary) solution for $(Q_2)$; see Table 2. This evidently raises the issue of the *sub-optimality gap* for SDR. Based on the results of Tan & Rasmussen (2001) and Luo et al. (2010), we show in Proposition 4 (see Appendix A.6) that SDR (Table 2) is optimal (asymptotically in $D$) in recovering the *binary solution* of $(Q_2)$.

We highlight that the performance bound in Proposition 4 compares the quality of both the approximate *binary solution* offered by SDR (with respect to the optimal binary solution of $(Q_2)$), as well as their respective cost functions. The asymptotic optimality is empirically validated in Appendix A.7.

## 3.3 Algorithm Description

The BCD-based algorithm alternates between refining the indicator and PMF vectors; see Algorithm 1. Lemma 2 (Appendix A.6) shows its convergence to a stationary point of $(Q)$.

---

**Algorithm 1** Iterative computation of KMs

---

// *Randomly Initialize* $\{\boldsymbol{\theta}_u^{(1)} \in \mathcal{P}\}$
**for** $n = 1, 2, ...$ **do**
    Compute $\boldsymbol{S}_i^{(n)}$ and $\boldsymbol{t}_i^{(n)}$ using (4)
    Call $\hat{\boldsymbol{\psi}}_i^{(n)} = \text{SDR}(\boldsymbol{S}_i^{(n)}, \boldsymbol{t}_i^{(n)}, M_{rnd}), \forall i \in \mathcal{I}_K$
    Compute $\boldsymbol{Q}_u^{(n)}$ and $\boldsymbol{r}_u^{(n)}$ using (3)
    // *Initialize FW with* $\{\boldsymbol{\theta}_u^{(n-1)}\}$, *from previous iteration*
    Call $\boldsymbol{\theta}_u^{(n)^\star} = \text{FW}(\boldsymbol{Q}_u^{(n)}, \boldsymbol{r}_u^{(n)}, \epsilon)$, for all $u \in \mathcal{U}_K$
**end for**

---

# 4 Interpretability via Causal Relation

## 4.1 Causal Relations

Once a solution is found using Algorithm 1, here, we propose a method to find causal relations among the RVs. More specifically, we compare the support set of each pair of RVs from the training set, $X_{u,i}$ and $X_{u,j}$, and check if there is any 'overlap' between their support set. Intuitively, this condition means that some of the elementary events (see Section 2.1) of one RV might be 'contained' in the elementary events of another. Consequently, the RVs are mutually related by causality, and the outcome of one determines that of the other. This insight is formalized here.

**Proposition 1 (Inclusion of Support Set)** *Consider two random variables $X_{u,i}$ and $X_{u,j}$ (belonging to the training set) whose KM are given by the model in* (1). *If* $\text{supp}(\boldsymbol{\psi}_j) \subseteq \text{supp}(\boldsymbol{\psi}_i)$, *then the following two causal relations hold:*

$$X_{u,i} = 1 \text{ implies } X_{u,j} = 1, \text{ and } X_{u,j} = 2 \text{ implies } X_{u,i} = 2. \tag{6}$$

Proof: See Authors (Oct 2017) for the proof.
Stated plainly, when the support set condition holds, the first outcome of $X_{u,i}$ implies the same outcome for $X_{u,j}$, and the second outcome for $X_{u,j}$ implies the second one for $X_{u,i}$, thereby implying a *mutual causal relation* among them (since $X_{u,i}$ influences $X_{u,j}$ and vice-versa). Note that our above definition of causality and causal relations is different than conventional ones in Pearl (2009)[Chap 2.8]. Moreover, our definition is distinct from Granger causality, due to the mutual coupling among the RVs in question.

We present next a special case of Proposition 1. When $\boldsymbol{\psi}_i = \mathbf{1}$, then $\text{supp}(\boldsymbol{\psi}_i) = \{D\}$, and $\text{supp}(\boldsymbol{\psi}_j) \subseteq \text{supp}(\boldsymbol{\psi}_i)$ holds, for any choice of $\boldsymbol{\psi}_j, \forall j \in \mathcal{I}_K$, where $\mathcal{I}_K$ defined in Section 3.

**Corollary 1 (Maximally Supported RVs)** *Let* $\{\boldsymbol{\psi}_i, \boldsymbol{\theta}_u\}_{(u,i)\in\mathcal{K}}$ *denote the KM associated with* $\{\mathbb{P}[X_{u,i} = 1]\}_{(u,i)\in\mathcal{K}}$. *We define $\mathcal{M}$ as set of RVs for which the support set of the indicator vector is one, i.e.,* $\mathcal{M} = \{i \mid \boldsymbol{\psi}_i = \mathbf{1}\}$. *Then, the condition* $\text{supp}(\boldsymbol{\psi}_j) \subseteq \text{supp}(\boldsymbol{\psi}_i)$ *(Proposition 1) holds trivially* $\forall j \in \mathcal{I}_K$. *Thus, the causal relations in* (6) *hold, for every* $i \in \mathcal{M}$.

For maximally supported RVs, the realization of one outcome, $X_{u,i} = 1$, determines that of *all RVs of the set* $\{X_{u,j} = 1 \mid \forall j \in \mathcal{I}_K\}$.

**Example 1 (Causal Relations in Recommendation Systems)** In addition to prediction, recommendation systems are designed to accurately mine for *association rules*: if a user likes item $i$ will he/she like another item $j$? Thus, the causal relations of Proposition 1 and Corollary 1 will be quite powerful, as we shall see. In the example of Section 2.2, note that $\text{supp}(\boldsymbol{\psi}_1) \subseteq \text{supp}(\boldsymbol{\psi}_2)$. Then, Proposition 1 yields: if user 1 (or user 2) likes movie 2 implies he/she also likes movie 1. Moreover, $X_{1,2}$ and $X_{2,2}$

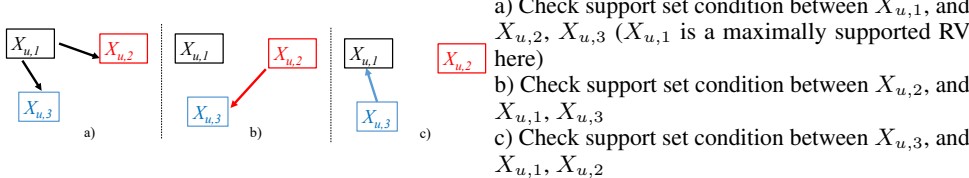

a) Check support set condition between $X_{u,1}$, and $X_{u,2}$, $X_{u,3}$ ($X_{u,1}$ is a maximally supported RV here)
b) Check support set condition between $X_{u,2}$, and $X_{u,1}$, $X_{u,3}$
c) Check support set condition between $X_{u,3}$, and $X_{u,1}$, $X_{u,2}$

Figure 1: Algorithm for CRM (toy example)

are maximally supported RVs, since $\psi_2 = 1$. Thus, Corollary 1 reads: If *any* user likes item 2, then this *causally implies* that he/she likes *all other items* in the training set.

Thus, our approach provides association rules that follow from *causal relations* between different RVs. Consequently, these relations are stricter (as they are rooted in Kolmogorov probability) than other methods for mining association rules, which are not based on causality.

## 4.2 CAUSAL RELATIONS MINING (CRM)

We provide an efficient algorithmic approach to automatically mine the above relations. In a nutshell, the algorithm does a pairwise check of the support set condition (Proposition 1), for pairs of RVs $X_{u,i}$ and $X_{u,j}$, $\forall (i,j) \in \mathcal{I}_K \times \mathcal{I}_K$ . The method is illustrated in Figure 1, for the illustrative example of Section 2.2. The above causal relations can be modeled conveniently using the *adjacency matrix* $A \in \mathbb{B}^{|\mathcal{I}_K| \times |\mathcal{I}_K|}$, defined as

$$[A]_{i,j} = 1, \text{ if } \mathrm{supp}(\psi_j) \subseteq \mathrm{supp}(\psi_i), \text{ and } 0 \text{ otherwise, } \forall i \neq j \ . \tag{7}$$

Stated differently, $[A]_{i,j} = 1$ if the support of $X_{u,j}$ is contained in that of $X_{u,i}$. Note that when $A$ is sparse (resp. dense) is an indication for a dataset in which few (resp. many) casual relations exist. To quantify the sparsity level, we define an influence score, $\beta_i = |\mathcal{I}_K|^{-1} \sum_{\substack{j \in \mathcal{I}_K \\ j \neq i}} [A]_{i,j}$ , which measures the normalized number of pairs $X_{u,i}$ and $X_{u,j}$, satisfying the support set condition. These steps are summarized in the Algorithm 2 (illustrated in Figure 1). Moreover, its complexity is dominated by the pairwise search over $\mathcal{I}_K$ (see Step 1), which comes at $\mathcal{O}(|\mathcal{I}_K|^2 - |\mathcal{I}_K|) \approx \mathcal{O}(|\mathcal{I}_K|^2)$ operations.

---

**Algorithm 2** Causal Relations Mining (CRM)

---

1. Check the support set condition, via a pairwise search to check for pairs $\psi_i$ and $\psi_j$ satisfying $\mathrm{supp}(\psi_j) \subseteq \mathrm{supp}(\psi_i)$, $\forall (i,j) \in \mathcal{I}_K \times \mathcal{I}_K$, $i \neq j$ (done over the training set $\mathcal{K}$)
2. Build the adjacency matrix $A$, in (7) , and compute the influence score $\beta_i$, $\forall i \in \mathcal{I}_K$
3. Find all pairs $(i,j)$ such that $a_{i,j} = 1$. For each of these pairs it holds that (Proposition 1), $X_{u,i}$ and $X_{u,j}$ are causally related, i.e.,

$$X_{u,i} = 1 \text{ implies } X_{u,j} = 1, \text{ and } X_{u,j} = 1 \text{ implies } X_{u,i} = 1 \tag{8}$$

4. Identify (if possible) maximally supported RVs (Corollary 1), $\mathcal{M} = \{i \mid \psi_i = 1\}$. For each of them, the relations in (8) hold *for all* $i \in \mathcal{I}_K$

---

## 5 SPECIAL CASES AND APPLICATIONS

We highlight relevant special cases and applications of our approach.

**Special Case: Unsupervised Learning Setting.** Note that Algorithm 1 is equally applicable to an unsupervised learning task (no prediction needed). Moreover, if the training set has no missing data, i.e., $\mathcal{K} = \mathcal{D}$, then an alternate solution to $(Q)$ may be obtained using the binary matrix factorization (MF) (Slawski et al., 2013) method. However, it *is not* applicable when factorizing a training set $\mathcal{K}$ (where $\mathcal{K} \subset \mathcal{D}$). Unlike Algorithm 1, binary MF does not offer prediction.

**Application: Analysis of Gene Expression.** The ability to analyze gene expression data is critical to DNA research (Zhang et al., 2009). This task can be formulated in our proposed framework as follows: $\mathbb{P}[X_{u,i} = 1]$ denotes the probability that the genes being studied are *expressed* in

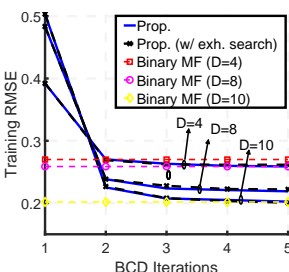 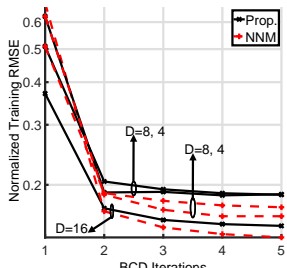 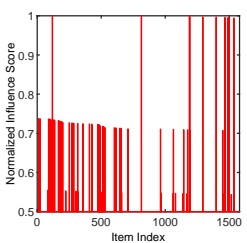

(a) Normalized Training RMSE vs Iterations (Setting 1)  (b) Normalized Training RMSE for KM vs NNM (Setting 2)  (c) Influence score $\beta_i$ ($D = 8$, Setting 2)

Figure 2: Performance of proposed method

sample $u \in \mathcal{U}$ at location $i \in \mathcal{I}$ of the DNA. In this setting, the set Kolmogorov elementary events, $\{\omega_1, \cdots, \omega_D\}$, represents the *various genes* that might be involved. Consequently, our approach models the *probability that a set of genes is expressed as a convex mixture of $D$ various genes*. We recall that this intuition follows from our model and is not enforced explicitly. More importantly, the causal relations can identify groups of DNA locations for which the expression of a gene (or its absence) in a given location implies its presence (or its absence) for all other locations in the group. We will numerically show the usefulness of these relations to DNA analysis, in Section 7.2.

## 6 PRACTICAL CONSIDERATIONS

**Overfitting:** Regularization parameters (to mitigate overfitting) can be included without any changes to the solution method. An $\ell_2$−regularization can be included in $(Q_1)$: $f(\boldsymbol{\theta}_u) = \boldsymbol{\theta}_u^T(\boldsymbol{Q}_u + \lambda_u \boldsymbol{I}_D)\boldsymbol{\theta}_u - 2\boldsymbol{\theta}_u^T \boldsymbol{r}_u + \gamma_u$ , where the regularizer $\lambda_u \geq 0$ is absorbed into a "new" matrix $(\boldsymbol{Q}_u + \lambda_u \boldsymbol{I}_D)$. Note that an $\ell_1$−regularization for $\boldsymbol{\theta}_u$ would not work, since $\boldsymbol{\theta}_u \in \mathcal{P}$. Similarly, an $\ell_1$-regularization for $(Q_2)$ is: $g(\boldsymbol{\psi}_i) = \boldsymbol{\psi}_i^T \boldsymbol{S}_i \boldsymbol{\psi}_i - 2(\boldsymbol{v}_i - (\mu_i/2)\mathbf{1})^T \boldsymbol{\psi}_i + \gamma_i$ , where the regularizer $\mu_i \geq 0$ is absorbed into the linear term, since $\mu_i\|\boldsymbol{\psi}_i\|_1 = \mu_i \mathbf{1}^T \boldsymbol{\psi}_i$, for $\boldsymbol{\psi}_i$ binary.

**Optimality Gap:** We recall that the proposed SDR method was shown to be quasi-optimal in providing approximate *binary solutions* to $(Q_2)$. Thus, the relaxation does not affect the interpretability, in the sense that Proposition 1 and Corollary 1 still hold. While the derivations pertaining to causal relations (Section 4) assume globally optimal solutions to $(Q)$ - an NP-hard problem, Algorithm 1 guarantees locally optimal ones. Thus, a bound on the gap between these solutions is needed. We highlight this issue as an interesting topic for further investigation.

### 6.1 LIMITATIONS

**Computational Complexity:** The computational complexity of Algorithm 1 is dominated by the solution in (5), which is $\approx \mathcal{O}(D^{4.5})$ operations per iteration of Algorithm 1 (recalling the negligible cost of the FW method). Notice that the additional complexity compared to matrix factorization (its extensions), for which the complexity is $\mathcal{O}(D^3)$ (e.g., $D \leq 16$ in all numerical results). Moreover, we are already investigating complexity reduction techniques leveraging the structure of the SDP, and distributed solutions to $(P)$ to enable *parallelization*. Finally, the computational complexity of Algorithm 2 consists mainly of step 1, which has $\approx \mathcal{O}(|\mathcal{I}_K|^2)$ operations.

**Learning a non-stationary distribution:** The proposed method assumes that distributions of the RVs in the training set are stationary: Indeed, scenarios with *time-varying distributions* are a limitation (and interesting future directions). However, in learning it is quite common to assume that the data-generating distribution is stationary.

## 7 NUMERICAL RESULTS

### 7.1 APPLICATION TO RECOMMENDATION SYSTEMS

**Experimental Setup:** We evaluate the performance of Algorithm 1 under the following. We opted to not have a stopping criterion based on the prediction error of the algorithm (on a validation set), since

we are primarily interested in the model that we learn in the training phase. Nonetheless, as we have reported in Appendix A.7, there is not loss in the predictive performance of the model. Hereafter, "Prop." refers to the proposed method.

Setting 1: An artificial training dataset, where $p_{u,i} \in \mathcal{K} = \{U = 20\} \times \{I = 40\}$, where $\{p_{u,i}\}$ are i.i.d. and uniformly chosen on the unit interval. We benchmark against a variant on Algorithm 1, where the $\psi$-step for $Q_2$ is replaced by an *exhaustive search*. As the data is artificial (unsupervised learning setting), we include the *Binary MF* in Slawski et al. (2013)[Algorithm 2].

Setting 2: The training set $\mathcal{K}$, is chosen as the MovieLens 100K, with $U = 943$ users and $I = 1682$ items, split into $80\%$ for training and $20\%$ for testing. Let $\{\hat{\boldsymbol{\psi}}_i\}, \{\hat{\boldsymbol{\theta}}_u\}$ the output of Algorithm 1, after 5 iterations. We benchmark against *matrix factorization (MF)* (Koren et al., 2009), *non-negative MF (NMF)* (Lee & Seung, 2001), *SVD++* (Koren, 2008) (ensuring the dimension of the factorization, $k$, is close to $D$), *non-negative models (NNM)* Stark (2015), and the $K$-*means (K-M)* algorithm. The implementation and results use the MyMediaLite package (Gantner et al., 2011).

**Training Performance for Unsupervised Learning (Setting 1):** Fig 2a shows the resulting normalized training RMSE $= (\sum_{(u,i) \in \mathcal{K}} |p_{u,i} - \hat{\boldsymbol{\theta}}_u^T \hat{\boldsymbol{\psi}}_i|^2 / |\mathcal{K}|)^{1/2}$. We observe that the monotone convergence in Lemma 2 is validated numerically, and that the training error decays with increasing model size, $D$. Note that the training performance of Algorithm 1 is indistinguishable from its exhaustive search variant. Moreover, Algorithm 1 converges to solution whose performance is similar to Binary MF, with a few iterations (except for $D = 8$ where Algorithm 1 outperforms Binary MF).

**Training Performance for Supervised Learning (Setting 2):** Recall that Binary MF is not applicable here, due to the supervised learning setting. The same conclusions hold when testing Algorithm 1 on the ML100K (Figure 2b). We underline that while NNMs yield better training performance over Algorithm 1, the latter will have better test performance (since NNMs are indeed defined by relaxing KMs). In Table 4 (Appendix A.7), we empirically verify that the test performance of the proposed method outperforms the benchmarks in Setting 2.

**Interpretability via Causal Relations (Setting 2):** We numerically evaluate the relations of Algorithm 2. The influence score for each item in the training set, $\beta_i$, is shown in Figure 2c where we displayed items with *'high' influence score*, $\beta_i \geq 0.5$. Note that each of these high influence items is causally related to at *least half* of the items in the training set. This confirms the effectiveness of Algorithm 2 for finding causal relations, and that a sparse adjacency matrix is uncommon. We next identify the set of items corresponding to maximally supported RVs, $\mathcal{M} = \{119, 814, 1188, 1190, 1290, 1393, 1462, 1486, 1494, 1530, 1590, 1638\}$. For each of these items, a user liking one given item, implies he/she likes all other items in the training set. Interestingly, these results remain the same when $D = 24$, thereby suggesting that procedure for mining causal relations is quite stable.

## 7.2 APPLICATION TO GENE EXPRESSION

**Experimental Setup:** Following the problem statement in Section 5, we show the usefulness Algorithm 2 for gene expression. We first define another setting.

Setting 3: We use the REGED0 dataset [3]. Element $(u, i)$ in the input matrix, $[\boldsymbol{L}]_{u,i}$ represents the *level of gene expression* for sample $u$ at location $i$ of the DNA, and is reported as integer between 0 and $L_{\max}$. Similarly to recommendation systems (Section 2.2), the training set is obtained as $p_{u,i} := [\boldsymbol{L}]_{u,i}/L_{\max}$. Thus, $p_{u,i}$ denotes the (empirical) probability that the gene is expressed in sample $u$ at location $i$. [4] After running Algorithm 1, we use Algorithm 2 to mine causal relations. As this is unsupervised learning setting, we include the binary MF (Slawski et al., 2013) method.

**Results:** While Fig 3a shows the training performance for several values of $D$, Figure 3b plots the corresponding influence score that is obtained with our method. Fig 3a reveals a huge gap ($\approx 3\times$ less) between the training error of Algorithm 1 and Binary MF (unlike Figure 2a where both algorithms yield similar performance). This drastic degradation in the performance of binary MF compared to the small artificial of Figure 2a may be attributed to increasing the data/problem size (though we were unable to empirically verify this claim). Thus, we opted to use the solution of Algorithm 1 as

---

[3] http://www.causality.inf.ethz.ch/data/REGED.html
[4] Finding better/different mappings from $[\mathbf{L}]_{u,i}$ to $p_{u,i}$ is an interesting direction, yet outside the scope of this our work.

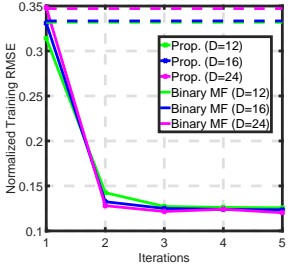

(a) Normalized Training RMSE as a function of iterations (Setting 3)

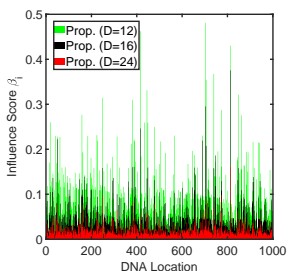

(b) Influence score for each DNA location (Setting 3)

Figure 3: Training performance of proposed method for REGED0 dataset

a basis for finding causal relations. We identified 10 DNA locations corresponding to the highest $\beta_i$ as: $\{813, 250, 774, 706, 380, 49, 477, 162, 740, 702\}$. For instance, the highest influence score of $.1612$ at DNA location $813$ allowed to identify a set, $\mathcal{S}$, of $161$ different locations which are *causally related* to DNA location $813$. More specifically, the expression (or absence) of a given gene in location $813$ implies its expression (or absence) in all other DNA locations in $\mathcal{S}$. While similar relations are possible using gene analysis methods, the above causal relations follow from our rigorous mathematical framework.

We also ran experiments on the ML 1M dataset (10x larger that ML 100K) and observed that main conclusions were the same. For lack of space, we instead opted to include different datasets such as the REGED0 dataset (5x larger than ML 100K) to exemplify another application of the approach. However, we note that a large-scale implementation of the method may be challenging at this stage. As mentioned earlier, we are currently investigating complexity reduction methods before running experiments involving large datasets. Rather, the current work is intended a proof of concept of the usefulness of such an approach.

# 8 CONCLUSION

We have proposed a framework for learning a Kolmogorov model, associated with a collection of binary random variables. Interpretability of the model (as defined by logical implication and causality) was harnessed by deriving causal relations, i.e., by finding sufficient conditions that bind outcomes of certain random variables. We also proposed an algorithm for computing a Kolmogorov model, a combinatorial non-convex problem, and showed its convergence to a stationary point of the problem using results from block-coordinate descent. The combinatorial nature of the problem was addressed using a semi-definite relaxation. We also proposed an efficient algorithm to mine for the causal relations inherent to our model. Our results suggest that increased interpretability and improved prediction, do not cause a significant increase in complexity. We highlight several key issues for future work, e.g., complexity reduction for (5) (leveraging that its dual is piece-wise linear), and a sufficient condition for identifiability (by adapting that of Fu et al. (2017))

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

# A   Supplementary Material for
# Learning Kolmogorov Models for Binary Random Variables

## A.1   Definitions

## A.2   Kolmogorov Model for a Random Variable

Let $(\Omega, \mathcal{M}, \mu)$ be a finite probability space. Here $\Omega$ is the sample space, the event class $\mathcal{M}$ is the power set of $\Omega$, and $\mu$ assigns probabilities to the sets in $\mathcal{M}$. Let $X_{u,i}$ denote a doubly-indexed set of random variables on the given probability space, having output alphabet in $\mathcal{A}$, where elements of $\mathcal{A}$ are indexed by $z$, i.e., $\mathcal{A}(z)$ represents the $z$th element in $\mathcal{A}$. Also, let $\mathbb{P}[X_{u,i} = \mathcal{A}(z)] \in [0, \ 1]$ denote the probability that outcome $\mathcal{A}(z)$ occurs, for $1 \leq z \leq |\mathcal{A}|$. We let $\Omega = \{\omega_d \mid 1 \leq d \leq D\}$, where $\{\omega_d\}$ the set of all $D$-elementary events. Since $X_{u,i}$ is binary, i.e., $\mathcal{A} = \{1, 2\}$, we write the KM for $X_{u,i}$ as,

$$\mathbb{P}[X_{u,i} = 1] = \boldsymbol{\theta}_u^T \boldsymbol{\psi}_{i,1}$$
$$\mathbb{P}[X_{u,i} = 2] = \boldsymbol{\theta}_u^T \boldsymbol{\psi}_{i,2} = 1 - \boldsymbol{\theta}_u^T \boldsymbol{\psi}_{i,1}, \tag{A.1}$$

where $\boldsymbol{\theta}_u$ is a *Probability Mass Function (PMF)* vector on the unit simplex, $\mathcal{P}$, and $\{\boldsymbol{\psi}_{i,1}, \boldsymbol{\psi}_{i,2}\} \in \mathbb{B}^D$ are *binary indicator vectors* representing the support of its probability measure. Moreover, the last equality follows from $\boldsymbol{\psi}_{i,1} + \boldsymbol{\psi}_{i,2} = \mathbf{1}$, which in turn follows from the outcomes of each RV summing to one. Since $X_{u,i}$ is binary, it is fully characterized by considering one outcome,

$$\mathbb{P}[X_{u,i} = 1] = \boldsymbol{\theta}_u^T \boldsymbol{\psi}_i, \tag{A.2}$$

## A.3   Related Work

**Matrix Factorization Methods:** Note that, $(Q)$ can be re-written as a low-rank matrix factorization problem, over the set of binary and stochastic matrices (see Appendix A.5. Thus, the proposed approach is connected to factorization methods: *Matrix Factorization (MF)* (Koren et al., 2009), *Nonnegative Matrix Factorization* (Lee & Seung, 2001), SVD (Cai et al., 2010) (and their many variants/extensions) have gained widespread applicability, covering areas in sound processing, (medical) image reconstruction, recommendation systems and prediction problems (Davenport & Romberg, 2016). These techniques model elements of the training set as inner product of two *arbitrary vectors*: Despite their success, the performance is inherently *tied to the validity of that model*, and consequently the extent to which these assumption hold. However, this inner product does not represent a RV (in a mathematical sense), when viewed in the context of the proposed model (see Section 2.1). Consequently, the analytical guarantees of Section 4, which underpin the causal relations, do not hold for general factorization methods.

**Nonnegative Sparse MF:** Hoyer (2004) numerically showed that the two low-rank components that are offered by sparse nonnegative MF (a variant of MF) provide insights on the data they model. While our proposed method also provides insight among the data (via the causal relations), the proposed KM and the resulting causal relations are based on established axioms in probability. Unlike the relations offered by nonnegative sparse MF which are shown empirically for a handful of examples, the causal relations (Section 4) hold analytically.

**Binary MF:** We underline that most MF methods also rely on BCD methods (alternating minimization), for which globally optimal solutions to each subproblem are needed for convergence. Thus, these methods cannot be *directly* applied to $(Q)$, due to the binary constraints on $\boldsymbol{\psi}_i$. The authors are unaware generic solution approaches for $(Q)$. Nonetheless, the *Binary MF* method (Slawski et al., 2013) can solve $(Q)$ when the problem is *feasible* and the training set spans the *entire* data set, i.e., $\mathcal{K} = \mathcal{D}$. However, the method operates in the *unsupervised learning* setting only, without providing prediction (see Section 5). This indeed limits the applicability of binary MF to practical scenarios, since real-world data will have missing/erroneous data.

**Clustering Methods:** Consider a special case of $(Q)$, where $\boldsymbol{\psi}_i$ is constrained to have one non-zero element. The resulting problem becomes the well-known $K$-*means clustering* (Lloyd, 2006). The K-means algorithms (and its variants K-medoids, fuzzy K-means and K-SVD), have become pervasive in an abundance of applications such as clustering, classification, image segmentation,

DNA analysis, online dictionary learning, source coding, etc. Our approach *generalizes K-means*, by allowing for overlapping clusters. While a similar generalization of the classical K-means algorithm was considered in (Whang et al., 2015), the number of points per cluster is determined explicitly.

**Nonnegative Models:** *Non-Negative Models (NNMs)* (Stark, 2016a) are recent attempts at interpretable models. For reasons of computational tractability (Stark, 2016a), NNMs are defined by relaxing $\psi_i$ in (1). However, this relaxation *impairs* the highly interpretable nature of the model in (1), making causal relations *less accurate*.

## A.4 Known context-dependent map

Consider the case where the elementary events correspond to each of the movie genres, i.e., $\Omega = \{\omega_1, \cdots, \omega_D\} = \{\text{"Action"}, \cdots, \text{"SciFi"}\}$. We refer to this as a *known context-dependent map*: $\mathbb{P}[X_{u,i} = 1] = \psi_i^T \theta_u = t_i^T \theta_u$ is expressed as a convex/stochastic mixture of movie genres. These models are a 'holy grail' for recommendation systems, due to the direct interpretation that they provide. In this case, $t_i$ is a binary genre tag vector with $[t_i]_m = 1, \forall\, m \in \{D\}$, if movie $i$ belongs to movie genre $m$, and zero otherwise. Now consider a genie-aided setting, where the set of movie genre tags $\{t_i\}$, are known (and consequently $\{\psi_i\}$ are known as well). Then, one can revisit the optimization problem for determining the KM, $(Q)$, where the indicator vectors are given, $\{\psi_i = t_i\}$, and the optimization is performed over the PMF vectors only $\{\theta_u\}$:

$$\min_{\theta_u \in \mathcal{P}} \mathcal{E}(\{t_i\}, \{\theta_u\}) \tag{A.3}$$

However, numerical result reveal that the training and test performance of these representations is quite poor. We tested the performance of this highly interpretable model, by extracting the tag vectors $\{t_i\}$ from the ML 100K dataset, and using them to optimize the corresponding PMF vectors. We set $D = 19$ to match the total number of movie genres for the ML 100K dataset. As seen in Table 3, the

| Training RMSE | Test RMSE |
|---|---|
| 0.4468 | 0.4468 |

Table 3: Normalized errors metrics when the context-dependent map is known (ML100K).

known context-dependent map have poor performance. This suggests the existence of a trade-off between the having this content-dependent map, and training/test performance.

## A.5 Problem Formulation in Matrix Form

Let $\Psi = [\psi_1, \cdots, \psi_I]$, $\Psi \in \mathbb{B}^{D \times I}$ denote the aggregate indicator matrix (containing all the individual indicator vectors), $\Theta = [\theta_1, \cdots, \theta_U]$, $\Theta \in \mathbb{R}_+^{D \times U}$ the aggregate matrix of PMF vectors, and $[P]_{(u,i)} = p_{(u,i)}, \forall (u,i) \in \mathcal{U} \times \mathcal{I}$, $P \in \mathbb{R}_+^{U \times I}$ the aggregate matrix of known probabilities. Then, $(Q)$ can be written in equivalent matrix form,

$$\begin{cases} \min_{\Psi, \Theta} \mathcal{E}(\Psi, \Theta) = \| M \circ (\Theta^T \Psi - P) \|_F^2 \\ \text{s. t.} \quad \Psi \in \mathbb{B}^{D \times I}, \ \Theta \in \mathbb{R}_+^{D \times U}, \ \Theta^T \mathbf{1} = \mathbf{1} \ . \end{cases} \tag{A.4}$$

where $\circ$ denotes the Hadamard product, and $M \in \mathbb{B}^{U \times I}$ is a *mask matrix* having $M_{u,i} = 1, \forall (u,i) \in \mathcal{K}$.

## A.6 Main Results

Below, we summarized the results used in the paper; see Authors (Oct 2017) for the proofs.

We use following known result to find the descent direction for the FW method (the proof is known).

**Proposition 2** *Consider the following Linear Program (LP),*

$$(P_{PS}) \ \ x^\star = \operatorname*{argmin}_{x \in \mathbb{R}^n} c^T x, \ \ \text{s. t.} \ \mathbf{1}^T x = 1, \ x \geq 0$$

*Its optimal solution is given by*

$$\boldsymbol{x}^\star = \boldsymbol{e}_{j^\star} \text{ , where } j^\star = \text{argmin}_{1 \le j \le n} \boldsymbol{c}^T \boldsymbol{e}_j$$

*Thus, the solution reduces to searching over the vector $\boldsymbol{c}$.* □

We show the convergence of the FW algorithm (Table 1).

**Proposition 3** *Let $\boldsymbol{\theta}_u^\star$ be the optimal solution to $(Q_1)$. Then the sequence of iterates $\{\boldsymbol{\theta}_u^{(k)}\}$ satisfies (Jaggi, 2013)[Theorem 1],*

$$\|f(\boldsymbol{\theta}_u^{(k+1)}) - f(\boldsymbol{\theta}_u^\star)\|_2 \le \mathcal{O}(1/k), \ k = 1, 2, \cdots \square$$

Proof: The linear convergence rate for all FW variants, was proved in Jaggi (2013)[Theorem 1].

**Quasi-optimality of SDR:** The question was studied extensively in the context of binary detection for multi-antenna communication (Tan & Rasmussen, 2001). Interestingly, $(Q_2)$ can be recast as a noiseless binary detection problem, where SDR has been to be optimal. The results is formalized below.

**Proposition 4** *Let $g(\boldsymbol{\psi}_i^\star)$ and $g(\hat{\boldsymbol{\psi}}_i)$ denote the optimal solutions to the binary QP in $(Q_2)$, and its SDR after randomization (Table 2), respectively. The approximation quality is defined as (Luo et al., 2010),*

$$\eta \le \ g(\boldsymbol{\psi}_i^\star)/g(\hat{\boldsymbol{\psi}}_i) \ \le 1. \tag{A.5}$$

*It holds that $\eta = 1$, with probability $1 - \exp^{-\mathcal{O}(D)}$, asymptotically in $D$. Thus, the relaxation is quasi-optimal.* □

Proof: See (Authors, Oct 2017).

**Lemma 2** *Let $t_n \triangleq \mathcal{E}(\{\boldsymbol{\psi}_i^{(n)}\}, \{\boldsymbol{\theta}_u^{(n)}\})$, $n = 1, 2, ...$ be the sequence of iterates, resulting from the updates in Algorithm 1. Then, $\{t_n\}$ is non-increasing in $n$, and converges to a stationary point of $(Q)$ in (2), almost surely.* □

Proof: The convergence is shown in (Authors, Oct 2017).

## A.7 ADDITIONAL NUMERICAL RESULTS

**Prediction Performance (Setting 2):** Since the range of the predicted variable is different for MF/NMF/SVD++, and KM/NNM, we use the normalized test RMSE, i.e., NRMSE $= \eta(\sum_{(u,i) \in \bar{\mathcal{K}}} |[\boldsymbol{R}]_{(u,i)} - \hat{R}_{u,i}|^2/|\bar{\mathcal{K}}|)^{1/2}$ where $\bar{\mathcal{K}}$ is the test set, and $\eta = (R_{\max} - R_{\min})^{-1} = 1/4$ is the normalization for MF/NMF/SVD++. For KMs/NNMs the same metric reduces to NRMSE $= \left( \sum_{(u,i) \in \bar{\mathcal{K}}} |[\boldsymbol{R}]_{(u,i)}/R_{\max} - \hat{\boldsymbol{\theta}}_u^T \hat{\boldsymbol{\psi}}_i|^2/|\bar{\mathcal{K}}| \right)^{1/2}$. The best values for $\lambda_u$ and $\mu_i$, were picked from a coarse two-dimensional grid by cross-validation, using a held-out validation set. The Normalized RMSE results are shown in Table 4. We observe a significant gap between KMs, and well known collaborative filtering methods, especially as $D$ increases. Moreover, the drop in performance for NNMs for increasing $D$ may be due to over-fitting.

**Asymptotic optimality of SDR for $\psi$-step solution:** Table 5 is a numerical validation of Proposition 4 where we computed the error rate of SDR (compared to the exhaustive search), aggregated over all iterations. We observe that the approximation error decreases, with increasing $D$ (following Proposition 4).

Table 4: Normalized Test RMSE (Setting 2). The dimension of factorization for MF/SVD++, $k$, is equal to $D$ (unless stated in the corresponding entry). The normalized test RMSE for all other methods are taken from the following repository: http://www.mymedialite.net/examples/datasets.html ('-' indicates the unavailability of the correspond test RMSE from the repository.

|       | $D = 4$ | $D = 8$        | $D = 16$        | $D = 24$         |
|-------|---------|----------------|-----------------|------------------|
| **KM**  | 0.199   | **0.2013**     | **0.1900**      | **0.1861**       |
| NNM   | **0.194** | 0.2255        | 0.2057          | 0.2118           |
| MF    | 0.229   | $0.228(k = 10)$ | $-$             | $0.226(k = 40)$  |
| SVD++ | 0.228   | $0.227(k = 10)$ | $0.227(k = 20)$ | $0.226(k = 50)$  |
| K-M   | 0.210   | .2096          | 0.2105          | 0.2105           |
| NMF   | $-$     | $-$            | $-$             | $0.192(k = 100)$ |

Table 5: Error rate for SDR (Artificial Dataset, $U = 20, I = 40$).

|                              | $D = 4$ | $D = 8$ | $D = 10$ |
|------------------------------|---------|---------|----------|
| SDR Accuracy $\times 10^{-3}$ | 7.5     | 4.4     | 4.0      |

