# OpenReview forum: "Learning Kolmogorov Models for Binary Random Variables"
_ICLR.cc/2019/Conference_

### Official Review · AnonReviewer3 · 2018-10-31
**the method is succinct and interesting, some concerns about data and complexity**

**Rating:** 8
**Confidence:** 4

**Review:**

The reviewer finds that the proposed method interesting. The model is very clean, and the implication in causal inference is significant. The writing is also clean and clear. The reviewer has several concerns:

1) the algorithm seems not very scalable. In the two subproblems, there is one solved by a large number of parallel SDRs. SDR is quite expensive, and for each column in the data matrix one has to solve an SDR in each iteration. This is too much for large scale recommender systems. In fact, in the experiment 1 on MovieLens, the algorithm was only tested on a not-so-large dataset and run 5 iterations. The reviewer feels that more scenarios should be tested (e.g., more iterations, various sizes of dataset, etc.). Fixing the number of iterations also sounds a bit funny since it is more intuitive to stop the algorithm using some validation set or when the algorithm converges under a certain criterion.

2) The algorithm works with *probability* of binary data. This is quite hard to estimate in practice. For example, people ``'likes'' a movie for only once. It is hard to tell what is the probability of generating this ````"like". It seems that the experiment part of this paper did not clearly state how to obtain the probability that the algorithm needs.

3) The proposed method is a special nonnegative matrix factorization, which could be unidentifiable. How to circumvent such situation? Since identifiability of NMF affects interpretability a lot.

---

> ### Author Response · Authors · 2018-11-23
> **Clarification to comments raised by AnonReviewer3**
>
> We are grateful for the positive feedback and constructive comments from the reviewer. We have prepared a revision addressing all the concerns, in which we have fundamentally rewritten many parts of the paper (highlighted in red). Moreover, kindly find below clarifications to the comments.
>
> 1)
> - "the algorithm seems not very scalable. In the two subproblems, there is one solved by a large number of parallel SDRs. SDR is quite expensive, and for each column in the data matrix one has to solve an SDR in each iteration. This is too much for large scale recommender systems"
> We thank the reviewer for pointing this issue. We have expanded the computational complexity discussion (Sec 6) to include the following:
> "The computational complexity of Algorithm~1 is dominated by the solution in~(5), which is ~ O(D^4,5)  operations per iteration of Algorithm~1 (recalling the negligible cost of the FW method).
> Notice that the additional complexity compared to matrix factorization (and its extensions), for which the complexity is  O(D^3) , is not significant, especially that the size of the factorization in recommendations systems, D, is small. We also recall that D is relatively small (D < 16 in all numerical results).
> Moreover, we are are already investigating complexity reduction techniques leveraging the structure of the SDP, and distributed solutions to (P) to enable parallelization. "
> More specifically, we will leverage the fact that strong duality holds for programs such as (5) to equivalently solve its dual:  due to the structure of the constraints (5) its dual is a piecewise linear program, which can be solved using a subgradient methods.
>
> - " In fact, in the experiment 1 on MovieLens, the algorithm was only tested on a not-so-large dataset and run 5 iterations. The reviewer feels that more scenarios should be tested (e.g., more iterations, various sizes of dataset, etc.)."
> We did run experiments on the ML 1M dataset (10x larger that ML 100K) and and observed that main conclusions were the same.  For lack of space, we instead opted to include different datasets such as the REGED0 (see sec 7.2) to illustrate the application of the our method to another research area. We should also underline that  REGED0  has ~ 500,000 samples, i.e., 5x larger than ML 100K.  However, we fully agree with the reviewer that a large scale implementation of the method may be challenging at this stage; As mentioned in our answer to your previous comment, we are currently investigation complexity reduction methods before running experiments involving large datasets. Rather, the current work is intended a proof of concept of the usefulness of such an approach.
> These clarification have been included in the revision.
>
> - "Fixing the number of iterations also sounds a bit funny since it is more intuitive to stop the algorithm using some validation set or when the algorithm converges under a certain criterion."
> We opted to not have a stopping criterion based on the prediction error of the algorithm (on a validation set), since, in this work, we are primarily interested in the model that we learn in the training phase: indeed, the focus of this work is to derive   causal relations, from the model that is learnt from the training set. Nonetheless, we included experiments on prediction error (Appendix A.6) to verify that there is not loss in the predictive performance of the model.
> Moreover, we also decided to fix the number of iterations to be same for all the different algorithms to ensure fairness among all the benchmarks. This discussion was added at the end of Sec 7.

---

> > ### Author Response · Authors · 2018-11-23
> > **Clarification to comments raised by AnonReviewer3 (continued)**
> >
> > 2) We thank the reviewer for pointing out this issue, which was also raised by AnonReviewer1 (comment 3). We have restarted our response here, for your convenience.
> > In the current version, we stated that p_u,i is obtained from the rating matrix for recommendation systems (sec 2.2), and the gene expression matrix for DNA expression analysis (sec 7.2). While explicit formula were given in an earlier version, we removed them for lack of space.
> > a) For the case of recommendation systems, we have remedied that by adding the following (Sec 2.2):
> > "The training set, consisting of an empirical probability that user $u$ likes item $i$, $p_{u,i}$, is obtained from the rating that user $u$ has provided for item $i$, i.e.,  $p_{u,i} = [R]_{u,i}/R_{\max} $ where $[R]_{u,i} \in \IN$ denotes the rating that user $u$ has provided for item $i$, and $R_{\max}$ the maximum  rating Stark 2015. For instance, if user $u$ rates item $i$ with a score of $[R]_{u,i}= 7$  (where the maximum rating is $10$), then the empirical probability that user $u$ likes item $i$ is $p_{u,i}=.7$." We note that the same method, to 'convert' ratings into probabilities, was done in Stark 2015.
> > b) Moreover,  we added a clarification on obtaining p_{u,i} in example of gene expression data  (sec 7.2):
> > "Element $(u,i)$ in the input matrix, $[L]_{u,i}$ represents the level of gene expression for sample $u$ at location $i$ of the DNA, and is reported as an integer between $0$ and $L_{\max}$. Similarly to recommendation systems (Section~2.2), the training set is obtained as $ p_{u,i} := [L]_{u,i}/L_{\max}  $. Thus, $ p_{u,i}$ denotes the (empirical) probability that the gene is expressed in sample $u$ at location $i$."
> >
> >
> > 3) We thank the reviewer for drawing our attention to this important issue. There are some works that derive sufficient conditions on for the identifiably of non-negative MF problems, e.g, Fu etal.  "on identifiability of non-negative matrix factorization". However, the problem setting for the optimization problem in these results is different from the one considered here: indeed, the constraint that one of the matrices in the factorization  is binary (see (2)) renders the sufficient conditions by Fu not applicable. However, we do agree with the reviewer that identifiability and conditions for the existence of a solution to (2) are very important issues for future research. It may be possible to prove such results by adapting the proof from these earlier results.
> > We have added this aspect as future work in the revised version of the conclusion, as part of the future work.

---

### Official Review · AnonReviewer2 · 2018-11-05
**interesting paper but somewhat unclear**

**Rating:** 5
**Confidence:** 2

**Review:**

The paper considers a solution to a statistical association problem. The proposed solution involves a decomposition they call a kolmogorov model (what sort is not justified in any way and confused me a lot). The decomposition has two parts 1) a discrete basis function that needs to be discovered and 2) a discrete distribution over the basis elements. The define an optimization problem (2) which has a data term and some binary and simplex constraints and they propose a relaxation and decomposition of this optimization problem. They go on to claim that (mutual) causal relations can be then inferred by inspecting the representations they have learnt but they give little details on how and what impacts this distinction has in practice.  This may be obvious to a subfield expert but it is not clear to me at all. The paper is locally consistent but I have trouble understanding the contribution and placing in the broad machine learning field.

I am not an expert in causality so I cannot evaluate the contribution but I can say that what interests me are section 2.2 and sec. 4-5. And they both require a lot better writing. 2.2 made things much more intuitive but i fail to see how the indicator variable annotations (action, scifi, etc.) can possibly come out of the data. I think this is an important point to support the interpretability claim. As for 4 I think there is room for intuition building there as well as limitations (e.g. what sort of inferences can be made and not etc.) Finally for 5 i find that very interesting but i find it difficult to have the right intuition about what the support condition means and how that helps in a practical setting.

pros:
- causality and interpretability are major directions of research
- seems like a valid contribution on an interesting problem
cons:
- the highlevel picture is relatively clear but i find important things very difficult to grasp
- the kolmogorov model definition i find confusing but i am not an expert in causality (the introduction should give some intuition about what that is and why it is a good idea).
- find it very hard to have a coherent picture of the limitations and assess the contributions of the paper.

---

> ### Author Response · Authors · 2018-11-22
> **Clarifications for issues raised by AnonReviewer2**
>
> We thank the reviewer for all the insightful comments.
> We have prepared a revision addressing all these concerns, in which we have fundamentally rewritten many parts of the paper (highlighted in red). Kindly find below clarifications to some of the issues that were mentioned in your review.
>
> - "They go on to claim that (mutual) causal relations can be then inferred by inspecting the representations they have learnt but they give little details on how and what impacts this distinction has in practice.  This may be obvious to a subfield expert but it is not clear to me at all."
> We thank the reviewer for this suggestion. To address this comment, we have modified System Model (Section 2) and Interpretability via Casual Relation (Section 4). In particular, we have added this descrption at the beginning of Sec 4.
> "After the training phase is over and a solution is found using Algorithm 1, here, we propose a method to find causal relations among the RVs. More specifically, we compare the support set of each pair of RVs from the training set, $X_{u,i} $ and $X_{u,j}$, and check if there is any "overlap" between their support set. Intuitively, this condition means that some of the elementary events (see Section 2.1) of one RV might be contained in the elementary events of another. Consequently, the RVs have mutual causality, and the outcome of one determines that of the other. "
>
> - " I have trouble understanding the contribution and placing in the broad machine learning field."
> Regarding the connection to existing related work, we wish to point the reviewer to the fact that Appendix A.3  includes an extended literature survey  (due to lack of space in the main text). We have also extended that appendix to better differentiate our proposal with the existing works. We kindly refer the reviewer to the revised version of  Appendix A.3, and the Proposed Approach paragraph (Sec 2.1).
>
> - ".. what interests me are section 2.2 and sec. 4-5. And they both require a lot better writing. 2.2 made things much more intuitive..."
> We are grateful to the reviewer for this observation on improving the clarity of Sec 4 and Sec 5, which we have done in the revised manuscript. We have added a few sentences at the beginning of each of them to provide an intuition for what follows.
>
> - "i fail to see how the indicator variable annotations (action, scifi, etc.) can possibly come out of the data. I think this is an important point to support the interpretability claim. "
>
> The approach consists of learning a (hidden) latent model by jointly optimizing the PMF and binary indicator vectors. Thus,
> interpreting elements of the indicator vectors as movie genres is not possible,  as it requires a context-dependent map elements of the binary vectors (obtained from the optimization problem)  to movie genres.  Another way to find such an interpretation is when  this context-dependent map is known apriori : In this setting, each item is already tagged with its movie genres, and the indicator vectors are thus known and need not be optimized. However,  simple numerical experiments reveal that the training/test performance is rather poor (see Appendix A.4).  This results points to a tradeoff between the having this context dependent map and the training/test performance. We should reiterate that in this work, interpretability is done via the causal relations.
> We have updated the exampled in Sec 2.2 to avoid possible misunderstandings, and added Appendix A.4.
>
>
> - "As for 4 I think there is room for intuition building there as well as limitations (e.g. what sort of inferences can be made and not etc.). Finally for 5 i find that very interesting but i find it difficult to have the right intuition about what the support condition means and how that helps in a practical setting"
> Regarding the intuition on the support set condition we refer the reviewer (again) to the discussion added at the beginning of Sec 4.1.
> We also refer the reviewer to Example 1 (Sec 4.1) to see the applications of the causal relations, to the illustrative example mentioned in Sec 2.2.
>
> - " the kolmogorov model definition i find confusing but i am not an expert in causality (the introduction should give some intuition about what that is and why it is a good idea)."
> We tried our best to improve/simplify the clarity of Sec 2.1 by putting all the analytical details in appendix A.1, and exemplifying the approach in Sec 2.2.
>
> - " I find it very hard to have a coherent picture of the limitations and assess the contributions of the paper."
> We have added a subsection in Sec 6 details the limitations of the approach (Sec 6) as the reviewer suggested.  Moreover, a paragraph was added to Sec 5 to briefly describe the scope of the possible applications.

---

> > ### Author Response · Authors · 2018-11-26
> > **Clarifications for issues raised by AnonReviewer2 (continued)**
> >
> >
> > - "The proposed solution involves a decomposition they call a kolmogorov model (what sort is not justified in any way and confused me a lot). "
> > The reason for naming (1) as a KM was simply because the decomposition model is based on the well-known Kolmogorov representation theorem (A. N. Kolmogorov  1961) . We have added some citations for interested readers.  But we would be grateful to reviewer if a more suitable nomenclature can be suggested.

---

### Official Review · AnonReviewer1 · 2018-11-07
**An interesting formulation and approach. Some concerns about pertinence and related work**

**Rating:** 5
**Confidence:** 4

**Review:**

The paper deals with an interesting problem. The presentation is clear the the approach intuitive.  However, the reviewer has some concerns about the pertinence of the approach and the relationship with related work.

It would be very helpful if the authors could contrast and compare the proposed approach (both qualitatively and quantitatively in their numerical experiments)  with methods for sparse non-negative matrix factorization. These would also lend themselves to causal interpretation.

The need for modeling probabilities p_u,i in the key motivating applications is questionable. Indeed in both recommendation systems and gene expression datasets the observations are not readily in the form of probabilities. For instance in the experiments, the authors normalize the level of gene expressions to make is look as a probability, which by the way is very different from the i.id uniform setup considered in setting 1. For the movielens dataset, it is unclear how the data was preprocessed to obtain the observed p_ui.

---

> ### Author Response · Authors · 2018-11-21
> **Clarifications for issues raised by  AnonReviewer1**
>
> We thank the reviewer for all the insightful comments that were raised.  We have prepared a revision addressing all these concerns, in which we have fundamentally rewritten many parts of the paper (highlighted in red).
>
> Regarding the connection to existing related work, we wish to point the reviewer to the fact that this was included in Appendix A.3 in the original submission (due to lack of space in the main text). We have also extended this discussion to emphasize the differences between these prior approaches and our proposed one. We refer the reviewer to the revised version of  Appendix A.3, and the Proposed Approach paragraph (Sec 2.1).
>
> Moreover, we have expanded the survey to include the works on sparse nonnegative Matrix Factorization (MF) that were suggested.  Namely, we have added the following qualitative discussion on the differences between this approach and our proposed one, to Appendix A.3:
> "Hoyer (2014) numerically showed that the two low-rank components that are offered by sparse nonnegative MF (a variant of MF) provide insights on the data they model. While our proposed method also provides insight among the data (via the causal relations), the proposed KM and the resulting causal relations are based on established axioms in probability.
> Unlike the relations offered by nonnegative sparse MF which are shown empirically for a handful of examples, the causal relation (sec 4) hold  analytically.  "
> We have also extended the discussion on differences to other related appreaches (see Proposed Approach in Sec 2.1).
> However, we were unable to find works where a causal interpretation is developed for spare non-negative  MF. We would  be grateful to the reviewer if he/she can point the work that he/she has in mind. This way, we may be able to obtain the quantitative results  for the work in question, in time for the camera ready version.
>
> We thank for raising the critical issue of obtaining p_u,i in practice: In the current version, we stated that p_u,i is obtained from the rating matrix for recommendation systems (sec 2.2), and the gene expression matrix for DNA expression analysis (sec 7.2). While explicit formula were given in an earlier version, we removed them for lack of space.
> 1)  In the revised version, we have remedied that by adding the following (Sec 2.2):
> "The training set, consisting of an empirical probability that user $u$ likes item $i$, $p_{u,i}$, is obtained from the rating that user $u$ has provided for item $i$, i.e.,  $p_{u,i} = [R]_{u,i}/R_{\max} $ where $[R]_{u,i} \in \IN$ denotes the rating that user $u$ has provided for item $i$, and $R_{\max}$ the maximum  rating Stark 2015. For instance, if user $u$ rates item $i$ with a score of $[R]_{u,i}= 7$  (where the maximum rating is $10$), then the empirical probability that user $u$ likes item $i$ is $p_{u,i}=.7$." We note that the same method, to 'convert' ratings into probabilities, was done in Stark 2015.
> 2) Moreover,  we added a clarification on obtaining p_{u,i} in example of gene expression data  (sec 7.2):
> "Element $(u,i)$ in the input matrix, $[L]_{u,i}$ represents the level of gene expression for sample $u$ at location $i$ of the DNA, and is reported as an integer between $0$ and $L_{\max}$. Similarly to recommendation systems (Section~2.2), the training set is obtained as $ p_{u,i} := [L]_{u,i}/L_{\max}  $. Thus, $ p_{u,i}$ denotes the (empirical) probability that the gene is expressed in sample $u$ at location $i$."
>
> We should underline that setting 1, having a training set of i.i.d. samples (Sec 7.1), was intended only as an  "artificial" example to empirically validate the convergence claim of Lemma 2.  The assumption that the training set consists of i.i.d. sample is not used anywhere else.

---

### Meta-Review · Area_Chair1 · 2018-12-14
**Clarification and comparison needed**

**Confidence:** 4
**Recommendation:** Reject

**Metareview:**

This work propose a method for learning a Kolmogorov model,  which is a binary random variable model that is very similar (or identical) to a matrix factorization model. The work proposes an alternative optimization approach that is again similar to matrix factorization approaches.  Unfortunately, no discussion or experiments are made to compare the  proposed problem and method with standard matrix factorization; without such comparison, it is unclear if the proposed is substantially new, or a reformation of a standard problem. The authors are encouraged to improve the draft to clarify the connection matrix factorization and standard factor models.